# Unraveling the Genomic Potential of the Thermophilic Bacterium *Anoxybacillus flavithermus* from an Antarctic Geothermal Environment

**DOI:** 10.3390/microorganisms10081673

**Published:** 2022-08-19

**Authors:** Júnia Schultz, Mariana Teixeira Dornelles Parise, Doglas Parise, Laenne G. Medeiros, Thiago J. Sousa, Rodrigo B. Kato, Ana Paula Trovatti Uetanabaro, Fabrício Araújo, Rommel Thiago Jucá Ramos, Siomar de Castro Soares, Bertram Brenig, Vasco Ariston de Carvalho Azevedo, Aristóteles Góes-Neto, Alexandre S. Rosado

**Affiliations:** 1Red Sea Research Center, Biological and Environmental Science and Engineering Division, King Abdullah University of Science and Technology, Makkah 23955, Saudi Arabia; 2Computational Bioscience Research Center, Biological and Environmental Science and Engineering Division, King Abdullah University of Science and Technology, Makkah 23955, Saudi Arabia; 3Institute of Biological Sciences, Federal University of Minas Gerais, Belo Horizonte 31270-901, MG, Brazil; 4Laboratory of Molecular Microbial Ecology, Institute of Microbiology, Federal University of Rio de Janeiro, Rio de Janeiro 21941-902, RJ, Brazil; 5Department of Biological Sciences, Universidade Estadual de Santa Cruz (UESC), Ilhéus 45662-900, BA, Brazil; 6Institute of Biological Sciences, Federal University of Pará, Belém 66075-110, PA, Brazil; 7Institute of Biological and Natural Sciences, Federal University of Triângulo Mineiro, Uberaba 38025-180, MG, Brazil; 8Department of Molecular Biology of Livestock, Institute of Veterinary Medicine, Georg August University, 37077 Göttingen, Germany; 9Bioscience Program, Biological and Environmental Science and Engineering Division, King Abdullah University of Science and Technology, Makkah 23955, Saudi Arabia

**Keywords:** *Anoxybacillus*, comparative genomics, phylogeny, potential metabolic functions, antarctica, polar volcano

## Abstract

Antarctica is a mosaic of extremes. It harbors active polar volcanoes, such as Deception Island, a marine stratovolcano having notable temperature gradients over very short distances, with the temperature reaching up to 100 °C near the fumaroles and subzero temperatures being noted in the glaciers. From the sediments of Deception Island, we isolated representatives of the genus *Anoxybacillus*, a widely spread genus that is mainly encountered in thermophilic environments. However, the phylogeny of this genus and its adaptive mechanisms in the geothermal sites of cold environments remain unknown. To the best of our knowledge, this is the first study to unravel the genomic features and provide insights into the phylogenomics and metabolic potential of members of the genus *Anoxybacillus* inhabiting the Antarctic thermophilic ecosystem. Here, we report the genome sequencing data of seven *A. flavithermus* strains isolated from two geothermal sites on Deception Island, Antarctic Peninsula. Their genomes were approximately 3.0 Mb in size, had a G + C ratio of 42%, and were predicted to encode 3500 proteins on average. We observed that the strains were phylogenomically closest to each other (Average Nucleotide Identity (ANI) > 98%) and to *A. flavithermus* (ANI 95%). In silico genomic analysis revealed 15 resistance and metabolic islands, as well as genes related to genome stabilization, DNA repair systems against UV radiation threats, temperature adaptation, heat- and cold-shock proteins (Csps), and resistance to alkaline conditions. Remarkably, glycosyl hydrolase enzyme-encoding genes, secondary metabolites, and prophage sequences were predicted, revealing metabolic and cellular capabilities for potential biotechnological applications.

## 1. Introduction

The genus *Anoxybacillus* belongs to the family Bacillaceae and the phylum Firmicutes. In contrast to the well-known genera *Bacillus* and *Geobacillus*, *Anoxybacillus* is a relatively new genus that was proposed in the year 2000 [1]. *A. flavithermus* was the first described species within this genus; it was formerly known as *Bacillus flavithermus* [2]. This species was discovered in a New Zealand hot spring and was characterized as a gram-positive, endospore-forming, facultative anaerobic microorganism, exhibiting intense yellow pigmentation and having the capacity to grow in a thermotolerant range (37–70 °C) [2]. Decades later, Pikuta et al. [1] isolated a new anaerobic strain (K1T) from animal manure. Based on its phenotypic characteristics (16S rRNA and DNA–DNA hybridization analyses), *Anoxybacillus* was proposed as a new genus in the family Bacillaceae, and the strain was named *A. pushchinensis* K1T. When both *A. pushchinensis* K1T and *B. flavithermus* were phylogenetically compared, they were found to be clustered together and distinct from *Bacillus.* Therefore, *B. flavithermus* was reclassified as *A. flavithermus* [1].

Since the introduction of high-throughput sequencing technologies, they have notably had a massive impact on genomic research activities, and whole-genome sequencing (WGS) is considered a valuable tool to improve the understanding of diversity, phylogenomics, and putative functionalities [3,4]. To date, a total of 29 *Anoxybacillus* species with validly published names have been reported; these species have been isolated from extremely diverse habitats, particularly geothermally heated environments. Of these, the genomes of only 18 species have been sequenced and deposited in public databases. So far, the complete genomes of only three *A. flavithermus* strains have been sequenced [5,6,7]. Analyses based on the limited number of *A. flavithermus* genomes have revealed that this bacterial species harbors genes that are compatible with its environmental adaptations, such as growth in high temperatures and alkalophilic conditions (e.g., pH higher than 8.5), with potential applications for catalysis [8,9].

Representatives of the genus *Anoxybacillus* are widely distributed and are mainly isolated from terrestrial hot springs and other thermophilic environments [8]. Therefore, information on *Anoxybacillus* species in geothermal environments from polar regions remains scarce. At present, only two studies have assessed the genus *Anoxybacillus* in cold environments. In particular, Poli et al. [10] described *A. amylolyticus* sp. nov., a thermophilic amylase-producing bacterium cultivated from the geothermal soil of Mount Rittmann volcano in Antarctica. Bendia et al. [11] isolated ten *Anoxybacillus* strains from the fumarole sediments of the Deception Island volcano (Antarctica) and identified all the strains as *A. kestanbolensis* via partial 16S rRNA sequencing. However, *A. flavithermus* in polar regions remains unexplored.

Therefore, more published *A. flavithermus* genomes are clearly needed to enhance the understanding of the genetic and potential metabolic diversity of this species, as well as the physiological mechanisms involved in its adaptation to geothermal environments. To date, the phylogenetic analysis of this genus has explored 16S rRNA genes and other gene markers. In this context, phylogenomic studies may expand the phylogenetic landscape of this important genus.

Studying the phylogenetic relationships and diversity of this interesting and recently described bacterial genus will not only provide taxonomic data but also serve as a useful route to completely unravel its biotechnological potential for applications in several industrial areas [12]. There is increasing interest in the genus *Anoxybacillus* for biotechnological applications, e.g., in biofuel, pharmacology, and the dairy industry, as well as in catalytic and bioengineering areas, mainly because of its capacity to produce thermostable enzymes and other metabolites with an economic value [13,14,15].

Here, we report the features, phylogenomics, and potential functionalities of seven *Anoxybacillus* strains isolated from the sediment samples from Deception Island—an active polar volcano in a marine system with continuous fumaroles, an accentuated gradient of temperature, salinity, and geochemistry located on the Antarctic Peninsula—based on an analysis of their genomes. To the best of our knowledge, this is the first study to deeply explore *A. flavithermus* species in the Antarctic geothermal environment.

## 2. Material and Methods

### 2.1. Knowledge Discovery in Databases: A Systematic Review of the Literature and Molecular Databases

The Scopus and PubMed databases are highly representative of scientific publications in the fields of microbiology and natural science [16]. Both these databases were checked in June 2021 to retrieve bibliographic records related to species belonging to the genus *Anoxybacillus* from 1982 (the first publication of the genus) to 2021. All the retrieved publications were analyzed to ensure consistency in the datasets and to avoid false records. To identify the relevant publications, the following search arguments were used in the combined fields of title, abstract, and keywords (per publication): (*Anoxybacillus*) AND (genom* OR metagenom* OR new species OR species).

In addition, using automated methods, all the available and public molecular databases were mined both at the individual sequence, genomic, and metagenomic levels to search for data associated with the genus *Anoxybacillus.* For this analysis, also conducted in June 2021, the following databases were used: PATRIC, NCBI, EBI, MG-RAST, IMG, and Microbiome Database. All files used for the systematic review are presented in Appendix A.

### 2.2. Sampling Site, Isolation, and Culture Conditions

Deception Island is characterized as a stratovolcano with a horseshoe shape and submerged caldera due to a strong eruption approximately 10,000 years ago [17]. Geothermal anomalies are observed in the volcano (e.g., Fumarole Bay, Whalers Bay, and Pendulum Cove), probably originating during the last eruptions between 1967 and 1970 [18]. Geothermal sites present extremely variable temperature and chemical conditions and are distributed in submerged and intertidal zones. In Fumarole Bay, temperatures range from 90 to 110 °C near the fumaroles, and from below zero to 50 °C in areas with less influence from fumaroles and closer to glaciers. Contrastingly, in Whalers Bay, the geothermal environments present temperatures between 40 and 60 °C, and in Pendulum Cove, 70 °C [19,20]. The gases emitted by Deception’s active fumaroles are mainly composed of carbon dioxide and hydrogen sulfide [21], and the pH surrounding the fumaroles ranges from 6.7 to 7.1 [22].

Sampling was performed during the XXXIV Brazilian Antarctic Expedition (between December 2015 and January 2016) at the geothermally active sites of Whalers Bay (−62.979611 N, −60.555778 W) and Fumarole Bay (−62.967417 N, −60.710111 W) on Deception Island, Antarctic Peninsula. Full characterization of the sampling points in the geothermal sites is described by Schultz et al. [15]. Approximately 500 g of sediment was collected from each sampling point at a depth of 0–10 cm. The samples were placed in sterile plastic bags and immediately transferred to the laboratory, where they were stored at 4 °C until they reached the Federal University of Rio de Janeiro, Brazil. Physicochemical parameters were assessed by Bendia et al. [22]. These include pH, electrical conductivity, granulometry, humidity, and micronutrient contents (B, Cu, Fe, Mn, and Zn), as well as organic matter, organic carbon, P, Si, Na, K, Ca, Mg, Al, sulfate, total nitrogen, nitrate, and ammonia contents.

To isolate thermophiles from the fumaroles in Deception Island, the samples were inoculated in six different culture media at a growth temperature of 55 °C, as previously described by Schultz et al. [15]. The chosen temperature was based on the in situ temperature of both the sampled geothermal sites, Fumarole Bay (ranging from 55–100 °C) and Whalers Bay (from 50–60 °C). The sediment samples were manually homogenized, and 10 g of the sample was added to 90 mL of saline solution (0.85%) with glass beads and agitated in an orbital shaker for 2 h. Serial 10-fold dilutions (10^−1^–10^−3^) were performed in the same diluent, and triplicates (0.1 mL of each dilution) were spread over the culture agar plates. The distinct media used for isolation were lysogeny agar [23], marine agar 2216 (Himedia), glucose yeast malt agar [24], DSMZ 260 medium (DSMZ GmbH), calcium phytate medium [25], and the National Botanical Research Institute’s phosphate growth medium (NBRIP medium) [26]. The plates were incubated at 55 °C for 48 h.

After incubation, colonies were selected based in their morphology for isolation. In total, 126 colonies were successfully isolated after purification and then selected for subsequent molecular analyses. Pure strains were stored at −80 °C in 20% *v*/*v* of glycerol and included in the Antarctic Culture Collection of Microbial Molecular Ecology Laboratory, Federal University of Rio de Janeiro, Brazil.

### 2.3. DNA Extraction and Genome Sequencing, Assembly, and Annotation

The strains were cultured in lysogeny broth for 48 h at 55 °C with constant shaking. Bacterial genomic DNA was extracted using the Wizard^®^ Genomic DNA Purification Kit (Promega, Madison, WI, USA), in accordance with the manufacturer’s instructions, and quantified using a Qubit (Invitrogen) fluorimeter.

Initially, the isolates were taxonomically identified on the basis of their 16S rRNA gene sequences. A 1450-bp-long fragment of the bacterial 16S rRNA gene was amplified using the primers 27F (5′-AGA-GTT-TGA-TCM-TGG-CTC-AG-3′) and 1492R (5′-TAC-GGY-TAC-CTT-GTT-ACG-ACT-T-3′) [27]. The PCR reaction mixture consisted of 10 pM of each primer, 10 ng of DNA, bovine serum albumin (1:20), 2× MyTaq^®^ reagent containing buffer Taq polymerase, dNTPs, and MgCl2 (Bioline, Taunton, MA, USA) in a final volume of 25 μL. Amplification was performed in a thermocycler, with initial denaturation at 95 °C for 3 min, followed by 25 cycles at 95 °C for 30 s, 55 °C for 30 s, and 72 °C for 30 s, and a final extension step at 72 °C for 5 min. The PCR products were purified using the SureClean Plus reagent (Bioline, Taunton, MA, USA). Purified amplicons were quantified using a Qubit 4.0 fluorometer with the Qubit^®®^ dsDNA HS Assay Kit (Life Technologies, Carlsbad, CA, USA) and examined via electrophoresis on 1.5% agarose gels. Approximately 25 ng of each amplicon was sent for sequencing using the Sanger’s chain termination technique [28].

Sequences were quality trimmed, primer sequences were clipped, and contigs were produced by merging overlaps from both forward and reverse sequences using Geneious Prime^®^. The assembled sequences were compared with all the DNA sequences deposited in the Silva v.132 database [29], considering a higher similarity between the obtained sequence and deposited sequences as a proxy for the molecular identification of the species. Seven bacterial strains isolated from both geothermal sites on Deception Island were identified as belonging to the genus *Anoxybacillus* (Table 1). Aiming to understand the functionalities and taxonomy of representatives of this thermophilic genus in cold environments, these seven strains were selected for WGS, the characterization of their genomes, and a comparison with other previously sequenced *Anoxybacillus* genomes.

In total, 5 μg/µL of gDNA was considered for the construction of paired-end sequencing libraries (2 × 150 bp) of a 450 bp insert size using the NEBNext^®^ Fast DNA Fragmentation and Library Preparation Kit (New England Biolabs Inc., Ipswich, MA, USA), according to the manufacturer’s instructions. The 2100 Bioanalyzer (Agilent Technologies, Santa Clara, CA, USA) was used for quality analysis of the final libraries. All the samples were sequenced on the Illumina Hi-Seq 2500 platform, in accordance with the manufacturer’s instructions.

The first step in the genome assembly process was to check the quality and remove the adapters and barcodes of the reads using FastQC v.0.11.5 [30] and Adapter Removal 2.3.0 [31]. The estimated best k-mers were selected using KmerStream 1.1 [32], followed by assembly using Edena v3.131018 [33] and SPAdes 3.14.1 [34]. In the next step, CD-HIT 4.8.1 [35] of the PSI-CD-HIT package was used to remove redundant contigs, producing a final file of contigs. Then, the quality of each assembly was checked in QUAST. For all the software, default parameters were applied. Bacterial genomes were annotated using Prokka v.1.14.6 [36], and BUSCO v. 4.0.1 was used to predict the completeness of single-copy orthologs [37].

The predicted contigs were also analyzed with GO FEAT [38], an online platform for the functional annotation and enrichment of genomic data integrated with the UNIPROT, INTERPRO, PFAM, NCBI, and KEGG databases. Annotation was performed using RAST [39], and the results were used to infer functions at subsystem levels. PCA was performed on the basis of the functional category data, in which the gene count data were previously transformed using a compositional data transformation (centered log ratio). BGCs were predicted using the web-based platform antiSMASH 5.0 [40]. The presence of bacteriocins was predicted using BAGEL 4.0 [41], and prophages were identified using PHASTER [42]. The web-based platforms ResFinder v. 4.1 [43] and VirulenceFinder v. 1.5 [44] were used to search for genes and chromosomal point mutations related to antimicrobial resistance and to detect the presence of virulence genes.

To predict genomic islands (GIs-metabolic islands, symbiotic islands, and virulence and resistance islands) and horizontally acquired regions, GIPSy was used [45]. As a reference genome for GIPsy, the genome of a nonvirulent strain (*G. stearothermophilus* DSM 458) was used. Moreover, Cas proteins and CRISPR-Cas system subtype classifications were predicted for *Anoxybacillus* strains using the software CRISPRcasIdentifier v. 1.1.0 [46].

### 2.4. Phylogenomic Analysis

To detect similarities among the seven *Anoxybacillus* genomes, Mash (fast genome and metagenome distance estimation) [47] was run using MinHash from the PATRIC platform with default parameters. JSpeciesW [48] with default parameters was used to calculate ANIb and TETRA values based on the seven *Anoxybacillus* genomes from the present study and the Bacillaceae genomes available in public databases. The same *Anoxybacillus* and Bacillaceae genomes were evaluated using Genome-to-Genome Distance Calculator (GGDC) 3.0, a digital DNA–DNA hybridization (dDDH) [49] provided by Leibniz on the DSMZ Institute website (https://ggdc.dsmz.de/ggdc.php#, accessed on 2 August 2022).

The R version 4.0.3 project and the Rstudio development environment were used to combine and analyze the ANIb and TETRA results, obtained from the web server. The relationships among the genomes were assessed on the basis of the *dist* function using Euclidean distance. Subsequently, the results were clustered using the *hclust* function with the average method. The ggplot2 package [50] was used to generate a scatter plot correlating the ANIb and TETRA values. The correlation between ANIb and TETRA values was assessed using the *cor.test* function with Spearman’s correlation, while the *shapiro.test* function was applied to evaluate the normality of the data, which indicated that the data did not follow a Gaussian distribution. Moreover, *pvclust* 2.2-0 [51] was used to calculate the confidence of each clade using the Euclidean and Ward.D methods, with a bootstrap value of 1000. A dendrogram was constructed using the factoextra package.

BUSCO was run for each genome, and BUSCO Phylogenomics was used to create the supermatrix. Subsequently, RAxML was used to construct the phylogenomic analysis, following the developer’s instructions [52]. The sequences of the supermatrix were aligned using Mafft [53], the best-aligned blocks were extracted using Gblocks [54], the Gblocks output was converted to Phylip format using ClustalW2 [55], and a phylogenomic tree was generated using RAxML [56].

### 2.5. Comparative Genomic Analyses

The comparison and annotation of orthologous gene clusters among the *Anoxybacillus* strains (n = 7) were performed in OrthoVenn2 (https://orthovenn2.bioinfotoolkits.net/home, accessed on 14 June 2021) [57], using the recommended parameters. Then, the orthologous groups were plotted and the number of common and exclusive protein orthologous groups present in the seven genomes were displayed.

BRIG [58] was used to provide a visual overview of the relationship between the genome sequences of the strains obtained in the present study and those available in public databases.

## 3. Results

### 3.1. Deep Mining of Anoxybacillus: Knowledge Discovery in Databases

Mining data on the genus *Anoxybacillus* were obtained from public databases (metagenomics projects and gene and protein deposits) and the available literature. In total, 78 *Anoxybacillus* genomes belonging to 18 species were retrieved from PATRIC (n = 73) and NCBI (n = 5). Although *Anoxybacillus* strains have been isolated from different locations around the globe, they have mainly been isolated from Germany (n = 25) and the Netherlands (n = 9) (Figure 1a). Strains have also been isolated from Russia, Antarctica, and Saudi Arabia. Remarkably, 34 genomes were mapped to *A. flavithermus* (Figure 1b), the most studied species in this genus. However, the complete genomes of only three strains have been sequenced to date [5,6,7].

Great effort was made to retrieve all the published data regarding the identification of new species in the genus *Anoxybacillus* and to improve the knowledge of this bacterial genus. In total, 25 scientific publications (all published in academic journals as research papers) from PubMed fit the search query for “*Anoxybacillus* AND (new species).”

Overall, 25 *Anoxybacillus* species were identified; these species were mainly isolated from hot springs or industries, such as milk powder factories (Appendix A). Different culture media were used to isolate *Anoxybacillus* species, with lysogeny broth (LB) and tryptic soy agar (TSA) being the most commonly used. Moreover, all the species and strains grew at a temperature range and pH range of 30–75 °C and 5.0–11.0, respectively. The optimal growth occurred under thermophilic conditions at temperatures varying from 45 to 60 °C, and the optimal pH was neutral pH. None of the strains were strictly anaerobic. Furthermore, in the case of species that had their whole genome sequenced, most genomes were draft genomes (Appendix A).

Both metagenomic studies that explored the genus *Anoxybacillus* in public databases detected it in only EBI and IMG. Conversely, the MG-RAST and Microbiome databases did not generate returns. Upon analyzing published studies that focused on metagenomics, seven accesses were retrieved (Appendix A). In brief, the microbial communities in the samples were accessed with metagenomic sequencing, and the genus *Anoxybacillus* was identified. The studies were mainly related to the assessment of the microbial structure and diversity in hot spring environments (n = 4).

### 3.2. Genome Features of Anoxybacillus Strains

De novo assembly of the genomes of seven *Anoxybacillus* strains generated many contigs, from 102 in LAT_31 to 1122 in LAT_38. The number of coding sequences (CDS) ranged from 2885 to 4297. Based on our findings and the data reported for the genome of *A. flavithermus* WK1 [6] (the only strain in the species with a complete genome), the genome size was around 3.0 Mb, varying between 2.7 and 3.5 Mb, with an average G + C ratio of 42% (41.6–43.5%). The general features of the draft genomes of *Anoxybacillus* strains are summarized in Table 2. CRISPR-Cas proteins were predicted in the *Anoxybacillus* strains LAT_11, LAT_26, LAT_27, LAT_31, LAT_33, LAT_35, and LAT_38, with at least one gene being detected in each genome. The CRISPR-Cas cassettes present in the genomes are classified in Table 3 (according to Makarova et al. [59]). There were three different system subtypes in strain LAT_27 (V-F, I-B, and III) and two in LAT_11 (V-F and I-U). LAT_38 displayed only the I-E system subtype.

### 3.3. Global Similarity and Supermatrix Phylogenomic Analyses

Phylogenomic analyses based on the complete distance matrix (ANI) and common subsequence frequency (TETRA) revealed that the *Anoxybacillus* strains present in our Antarctic thermophilic samples were very similar to each other (ANIb value > 98% and dDDH value > 82%). Based on the species used for comparison, the strains were phylogenomically closest to *A. flavithermus* (ANIb value, ~95% and dDDH value, > 70%). When the strains were compared with other representatives of the genus *Anoxybacillus* (*A. amylolyticus* and *Anoxybacillus* sp.), the ANIb values were lower than 80% and the dDDH was lower than 18%. The global similarity phylogenomic distance matrix (Figure 2a) revealed that all the newly sequenced *Anoxybacillus* strains were grouped with *A. flavithermus* WK1 upon applying the combined method based on ANIb and TETRA values. The results obtained from the ANIb and TETRA analysis from the whole genome data were used to calculate the distance matrix. Strain LAT_11 was grouped with *A. flavithermus* WK1 in the same clade, while the other strains were grouped in a separate clade. A supermatrix phylogenomic analysis (Figure 2b) revealed that the novel *Anoxybacillus* strains identified in the present study were grouped with each other with 100% certainty, in addition to grouping with *A. flavithermus* WK1. Thus, of the *Anoxybacillus* spp. present in the supermatrix phylogenomic tree, our strains were more similar to *A. flavithermus*. Moreover, a clear separation was noted between our strains and the *Brevibacillus* and *Geobacillus* strains, which were grouped in different clades. In addition, upon comparing the genomes of our *Anoxybacillus* strains with the genome of *A. flavithermus* WK1, a reference strain, differences were detected between the strains, mainly in the case of LAT_31, LAT_33, and LAT_35.

OrthoVenn2 was used to compare the orthologous clusters of the seven *Anoxybacillus* strains. The analyzed species formed 2976 clusters and 1386 single-copy gene clusters. The genomes of LAT_26 and LAT_38 shared 189 orthologous clusters, and those of LAT_11, LAT_27, LAT_31, LAT_33, and LAT_35 shared 113 orthologous clusters, indicating a high number of conserved (shared) genes among these genomes.

The image generated using BLAST Ring Image Generator, presented in Figure 3, also displays the genomic islands (GIs) found in the analyzed genomes. In total, 15 resistance and metabolic islands were predicted using Genomic Island Prediction Software (GIPSy) with a strong relationship.

### 3.4. Potential Functions of Thermophilic A. flavithermus Strains in Cold Environments

To compare the functional distribution between the seven *Anoxybacillus* strains, Rapid Annotation using Subsystem Technology (RAST) subsystems (Figure 4a) and GO FEAT outputs were assessed. The functional groups, with a significantly higher number of genes, were assigned to the subsystem amino acids and their derivatives, followed by the subsystem’s carbohydrates, protein metabolism, cofactors, vitamins, prosthetic group, and pigments. Interestingly, strain LAT_26 exhibited the highest number of genes in each functional category, while LAT_31, LAT_33, and LAT_35 exhibited a lower number of genes than the other strains.

Several encoding genes involved in the carbohydrate metabolism were revealed by GO FEAT in all the analyzed genomes. We focused on the analysis of glycosyl hydrolase enzymes. Genes related to α- and β-glucosidase, amylopullulanase, pullulanase, α-amylase, cyclomaltodextrinase, oligo-1,4-1,6-α-glucosidase, lytic murein transglycosylase, sucrase-6-phosphate hydrolase, α-galactosidase, α-L-arabinofuranosidase, sugar hydrolase/phosphorylase, and β-xylosidase were retrieved.

The seven *Anoxybacillus* strains had a similar number of genes related to potassium and sulfur metabolism, iron acquisition and metabolism, and aromatic compound metabolism. Genes related to stress response were found in all the analyzed *A. flavithermus* strains; however, genes related to oxidative stress and detoxification were in higher number in LAT_26, and the number of genes related to osmotic and periplasmic stress was greater in LAT_27.

Furthermore, prophage sequences within the genomes of the newly identified *Anoxybacillus* strains were retrieved. In total, 19 prophage regions, including 5 intact and 14 incomplete ones, were detected in all the genomes (Appendix A). These findings complemented those of the RAST annotation (Figure 4), highlighting that the genome of LAT_38 displayed a greater number of intact prophages with regions related to the prophages of *Thermus* and *Geobacillus* genera. In addition, prophage sequences from LAT_26 were identified with a G + C content of 41.44%, similar to that of the host genome (42%).

Based on the principal component analysis (PCA) plot (PC1 = 76.3%, PC2 = 17.6%), the *Anoxybacillus* strains LAT_31, LAT_33, and LAT_35 had a similar functional pattern and were clustered together (Figure 4b). Furthermore, strain LAT_38 formed a cluster alone and displayed the phage and prophage group as the main variable driving this pattern, corroborating with the prophage identification results using PHAge Search Tool-Enhanced Release (PHASTER) (Appendix A). The PCA results were similar to those of the phylogenomic analysis presented in Figure 2.

Genome annotation also revealed genes related to genome stabilization, DNA repair systems, adaptation to high temperature, and alkaline conditions. The data were manually examined in GO FEAT outputs. The list of detected genes with the corresponding locus tags is available in Appendix A. Regarding the adaptation of the strains to thermal stress in a geothermal site in Antarctica, all the genomes exhibited the chaperones involved in repairing heat-induced protein damage, DnaK and DnaJ, which are the co-chaperones for DnaK–GrpE interactions, as well as small Hsps (Hsp20), the gene *htp*X, and the ClpC and its related ClpP. *yfl*T and *hcr*A, genes involved in the heat adaptation, were not detected in the analyzed genomes. The genes related to GroEL were not detected; however, the genes associated with GroES were detected in LAT_31, LAT_33, and LAT_35. Two *csp* genes (*CspB* and *CspD*) and encoding Csps (cold-shock proteins) were also detected in all the genomes, potentially present to counteract the effect of a temperature decrease. Although the sediment in the Antarctic geothermal environment had an approximately neutral pH (between 6.5 and 7.5), genes involved in adaptation to alkaline pH, including those related to the Na^+^/H^+^ antiporter subunits A–G, were detected in all the genomes. The antiporter NhaC, also involved in alkalophilic adaptations, was not detected in any genome.

All the strains exhibited gene sequences involved in DNA repair mechanisms against environmental stresses, such as those related to enzymes with phosphodiesterase activity; DNA glycosylases (formamidopyrimidine–DNA glycosylase, A/G-specific adenine glycosylase, uracil–DNA glycosylase, and 3-methyladenine DNA glycosylase), as well as UvrA, UvrB, and UVrC for nucleotide excision repair. Genes encoding pythoene dehydrogenase, an enzyme related to carotenoid biosynthesis, were also detected, while those encoding pythoene synthase were not. *MutS*, *MutL*, and *LigA*, related to mismatch repair, were present in all the genomes. Similarly, *RecR* and *RuV*, related to homologous recombination repair, were present in all the genomes.

Secondary metabolites were also predicted and analyzed. The classes of genes most frequently involved in secondary metabolite production were polyketide synthase (PKS) and non-ribosomal peptide synthetase (NRPS). All the strains contained 16 biosynthetic gene clusters (BGCs) encoding several putative specialized molecules, including betalactone-NRPS, NRPS, and T3PKS (Appendix A). PRISM 4 revealed the presence of BGCs encoding nonidentified, non-ribosomal peptides in all seven strains. Moreover, BAGEL revealed the presence of sactipeptide, a member of the ribosomally synthesized and post-translationally modified peptides (RiPPs), in all seven strains.

Horizontally acquired resistance genes were examined using assembled contigs in the ResFinder database and GIPSy. No antibiotic resistance genes and no virulence genes were detected using ResFinder. Resistance islands were detected in the strains LAT_26, LAT_31, LAT_33, and LAT_35.

## 4. Discussion

Microorganisms thriving in geothermal environments have been the subject of extensive research, and members of the family Bacillaceae have been isolated in different contexts. *Anoxybacillus* species are widespread in manure and in the dairy industry, and most of them have been isolated from geothermal springs [9]. However, the presence of *Anoxybacillus* species in geothermal environments in polar regions is poorly explored. *A. amylolyticus* is the only novel species isolated from Antarctic geothermal soil [60]. To the best of our knowledge, the present study is the first to report finding *A. flavithermus* in Antarctica. Efforts have been made to elucidate the ecology and genomic features of *A. flavithermus* in the past decades; however, this species remains scarcely studied.

At present, several genomes of *Anoxybacillus* strains have been sequenced, providing genetic information regarding this thermophilic bacterium. However, only the strain WK1 of *A. flavithermus* has been completely sequenced [6]. Thus, a more in-depth analysis of the genomes is necessary to characterize the bacterial species. We performed, for the first time, phylogenomic analyses of the genus *Anoxybacillus*, including not only global similarity methods (ANI and TETRA) but also a supermatrix (concatenation) method using all single-copy orthologous proteins. Previous studies have revealed that *Anoxybacillus*, *Brevibacillus*, and *Geobacillus* are closely related to each other. Our phylogenomic analysis revealed *Brevibacillus* to be the sister group of *Anoxybacillus* and *Geobacillus* to be the sister group of the clade (*Anoxybacillus* + *Brevibacillus*). So far, previous small-scale (16S rRNA gene and beta-glucosidase) phylogenetic studies have reached no consensus regarding sister group relationships in the Bacillaceae family. Such studies are listed in Appendix A. Thus, to understand these relationships further, large-scale phylogenomic studies are still very necessary.

In the present study, we analyzed, for the first time, the presence, relative location, and gene composition of GIs in the bacterial species *A. flavithermus*. GIs are large blocks of genes that are acquired through lateral gene transfer and usually share a common group of related functionalities [45]. The seven lineages of thermophilic Antarctic *A. flavithermus* displayed metabolic, resistance, symbiotic, and virulence GIs, composed of metabolism-related, antibiotic-resistance-related, symbiosis-related, and pathogenicity-related genes, respectively. The total gene number and gene composition of these *A. flavithermus* lineages were generally distinct and may reflect the genome plasticity of the pangenome of this bacterial species [61], at least in the studied geothermic environment.

The presence of prophage sequences in *Anoxybacillus* was reported in detail by Saw et al. [6] and Goh et al. [8]. The authors observed intact prophage sequences in *Anoxybacillus* spp. SK3-4 and DT3-1, *A. kamchatkensis* G10, and *A. flavithermus* WK1 but not in *A. flavithermus* TNO-09.006, as well as incomplete prophages. In line with this finding, prophages were retrieved in all the analyzed genomes, particularly that of strain LAT_38. Prophages can be beneficial to their hosts, influencing bacterial fitness and lifestyle and conferring protection [62]. According to the existing analysis and literature, most of the probable HGT genes belong to the genus *Geobacillus*.

CRISPR (clustered regularly interspaced short palindromic repeats) and CRISPR-associated (Cas) proteins are defense systems present in microorganisms and act against phages and other foreign genetic elements [63]. The complex of Cas proteins (CRISPR-Cas) uses Cas proteins linked to CRISPR RNA (crRNA) [64,65]. This complex is directed to the proper sequence present in the invading DNA, usually through the recognition of the Protospacer Adjacent Motif (PAM) domain in the complementary strand of crRNA [63]. We found genes belonging to class 1 (type I and III). Type I is the most widespread type in CRISPR systems [59]. In almost all type I systems, pre-crRNA is processed by an RNase of the Cas6 family. Type III has a peculiarity: It is divided into two subtypes. CRISPR type III-A is capable of acting on plasmid DNA in vivo, while CRISPR type III-B is responsible for the cleavage of only single-stranded RNA in vitro. We also found one gene belonging to class 2 (type V). Type V is distinguished by a single RNA-guided RuvC-domain-containing effector, Cas12. However, additional studies are required to assess whether the predicted cassettes are rare subtypes of CRISPR-Cas systems and whether they are functional.

### 4.1. Adaptations to the Antarctic Geothermal Environments

The species *A. flavithermus* can flourish under thermophilic conditions and develop different approaches to survive in extreme environments, conferring high temperature tolerance. DNA-stabilizing mechanisms, efficient DNA repair systems, and adapted proteins are strategies required to thrive under harsh thermal conditions [66,67], and our strains showed all these modifications. Genes related to polyamine biosynthesis were detected in all the genomes; these genes play key roles in stress response, cell growth, and cell proliferation [68]. Distinct enzymes related to base excision repair were detected in the genomes, such as the DNA glycosylases; they are involved in the process of maintaining the integrity of the genomes, signalizing and removing specific damaged DNA bases, and filling the gap through DNA synthesis [69]. In addition, the genomes contained enzyme-encoding genes involved in carotenoid production and photoreactivation DNA repair, pathways that are important to protect the cells and their components against long-term UV exposure and to repair/remove damaged materials.

Another reason why our *A. flavithermus* strains could thrive in polar geothermal environments is the presence of mechanisms related to thermal adaptation. The analyzed strains contained various genes involved in heat- and cold-shock. These proteins have been repeatedly reported to be key players in protein folding and refolding. Furthermore, they can counteract some harmful effects of temperature downshift, thereby helping the cells to adapt [70]. They are also crucial in preventing the negative effects of temperature fluctuations, which are common in Antarctic geothermal sites, where the temperature can drastically change along small distances [22]. As anticipated, cold adaptation and cryoprotection were conferred by the genes encoding Csps. The synthesis of Csps is induced by the exposure of microorganisms to low-temperature conditions, and their presence in thermophilic genomes is very limited. Nevertheless, von König et al. [71] characterized a Csp from *Thermotoga maritima*, a hyperthermophilic bacterium. Nelson et al. [72] verified the presence of Csp homologs in the same bacterial species and in the thermophilic bacterium *B. caldolyticus*.

In response to harsh conditions, some microbial groups adopt reversible metabolic strategies to survive environmental stressors. Dormancy is one of the general strategies adopted to increase the cellular resistance to external stresses and to reduce energy expenditure; it allows bacteria to prosper after a disturbance caused by the environment. It is well known that *Anoxybacillus* species form spores as a survival strategy. In *Anoxybacillus*, the genome size is small, and the number of total genes involved in sporulation is usually less than 100 [73]; however, in LAT_26, 150 genes were detected.

### 4.2. A. Flavithermus as Potential Candidate for Biotechnological Applications

Apart from the biotechnological advances using nonconventional bacterial hosts as providers of bioproducts for biotechnological purposes, the use of the genus *Anoxybacillus* is still in its infancy. Here, we highlight the potential of the metabolic and cellular capabilities of *Anoxybacillus* strains as providers of bioproducts for bioconversion and bioengineering. *Anoxybacillus* species are common spoilage organisms in the food industry, particularly in dairy production plants. Therefore, most of the available information on this species is related to the dairy industry [7,74,75].

*A. flavithermus* is a thermophilic bacterium and displays genetic information for thermostable proteins and other bioproducts that make it a suitable possible candidate for deeper studies regarding their potential applications in biotechnology and astrobiology. Genomic analyses have revealed that most protein-encoding gene sequences are related to amino acids and derivatives and carbohydrate subsystems. Similar findings have been reported by Saw et al. [6] in *A. flavithermus* WK1 and by Goh et al. [8] in *Anoxybacillus* sp. SK3-4 and sp. DT3-1. We performed an in-depth analysis of the glycosyl hydrolase enzymes in the thermophilic *A. flavithermus* strains and detected the presence of genes encoding enzymes of industrial interest. Furthermore, we identified enzymes associated with two major biotechnological applications in the analyzed genomes: lignocellulose- and starch-related industries.

Enzymes capable of performing hydrolysis under extreme conditions of high temperature and pH are considered high-value bioproducts for the biofuel industry [76]. Our strains exhibited genes related to adaptation to thermal fluctuation and alkaline pH, as well as for hydrolases, key enzymes involved in the hydrolysis of lignocellulosic materials (e.g., β-xylosidase, α-L-arabinofuranosidase, and β-glucosidase). For instance, Chan et al. [77] isolated and characterized thermostable glucose-tolerant β-glucosidase from *Anoxybacillus* sp. DT3-1; it showed optimum activity at 70 °C and pH 8.5. As reported by Goh et al. [78] and Margaryan et al. [9], *Anoxybacillus* strains have the potential to provide enzymes that can be used in starch-related industries. Accordingly, genes encoding amylopullulanase, pullulanase, α-amylase, cyclomaltodextrinase, and oligo-1,4-1,6-α-glucosidase were identified in the genomes of *A. flavithermus* strains in the present study. α-amylases have been largely detected in *Anoxybacillus* strains, such as *A. flavithermus* DSM 2641 [79], *A. flavithermus* SO-13 [80], *Anoxybacillus* sp. SK3-4, and *Anoxybacillus* sp. DT3-1 [8]. Moreover, Ozdemir et al. [80,81] identified a thermostable α-amylase using RH as the substrate. The enzyme was obtained from thermophilic *A. flavithermus* SO-13 and showed optimum activity at 80 °C and pH 5.0. More studies are required to confirm these abilities under laboratory conditions.

An interesting possible application of *A. flavithermus* strains is in astrobiology due to the presence of several genes related to genome stabilization, adaptation to temperature shifts and alkalinity, spore formation, and carotenoid production, contributing to the survival of *A. flavithermus* under hostile terrestrial or extraterrestrial environmental conditions [6,8].

## 5. Conclusions

In conclusion, *A. flavithermus* strains are representatives of a new and underexplored group, widely isolated from thermophilic environments. However, little is known about their presence and adaptations to thrive in geothermal ecosystems in polar regions, as well as the use of the genomic approach to assess their genetic and metabolic capabilities.

Using genomic analysis, the present study revealed, for the first time, the lifestyle of the relatively new genus *Anoxybacillus*, its adaptations to overcome the harsh environmental conditions of Antarctica, and its bioproducts, which can have potential biotechnological applications. Moreover, based on the genomes of the Antarctic strains, we suggest more studies for deeper characterization and investigation regarding their applications in starch- and lignocellulose-related industries, as well as their possible applications in astrobiology, tailoring a novel direction for this thermophilic species.

## Figures and Tables

**Figure 1 microorganisms-10-01673-f001:**
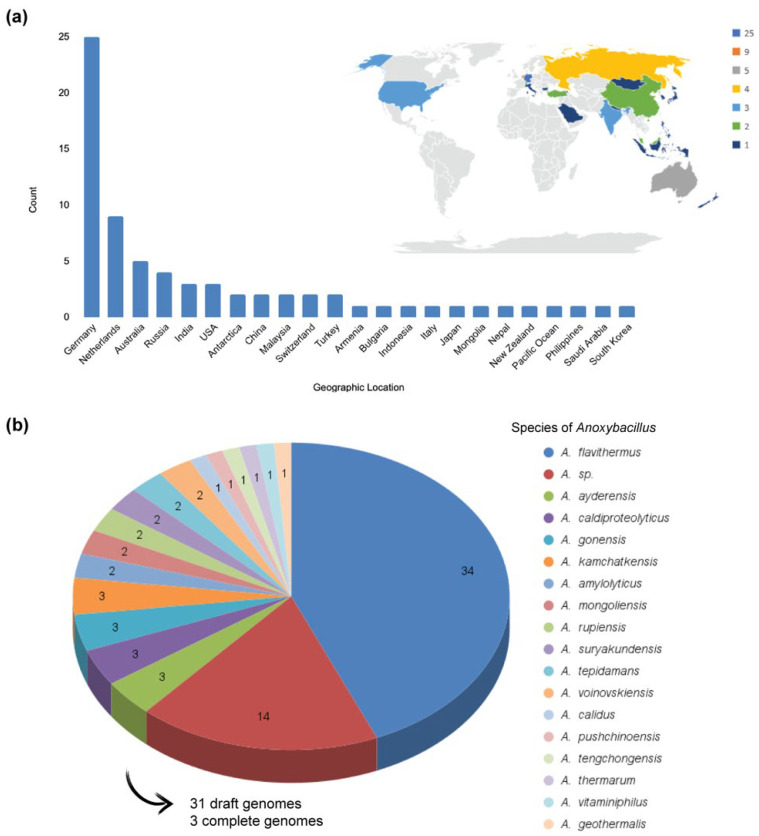
Genomic and metagenomic data on *Anoxybacillus* available in public databases. (**a**) Geographical distribution of *Anoxybacillus* strains. (**b**) Described species belonging to *Anoxybacillus* (n = 18) and the number of sequenced genomes in each species (draft and complete genomes, n = 34). Data were generated by PATRIC.

**Figure 2 microorganisms-10-01673-f002:**
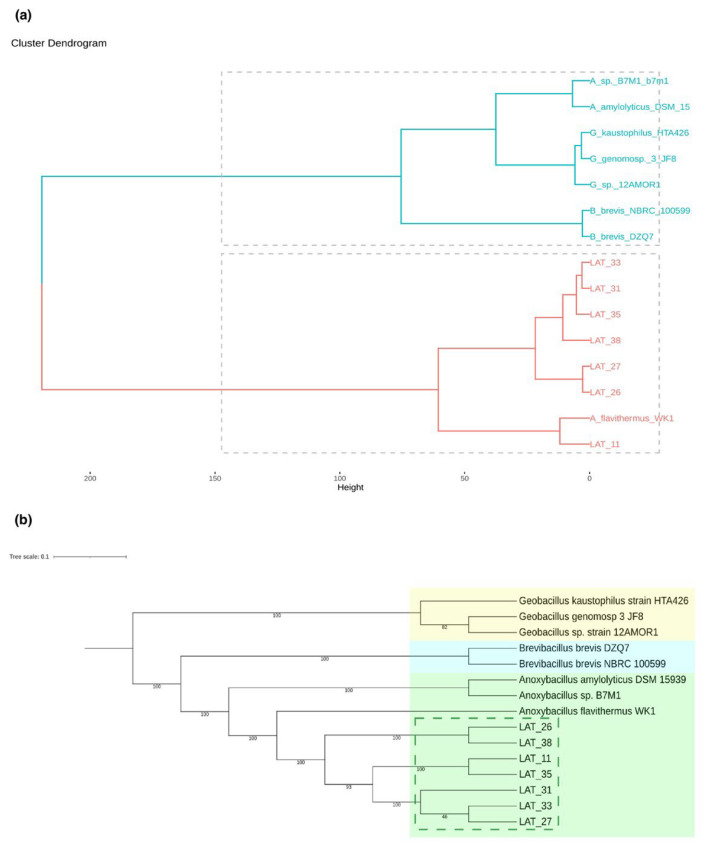
(**a**) Global similarity phylogenomic tree (based on ANIb and TETRA results) consisting of *Brevibacillus* (B_), *Geobacillus* (G_), *Anoxybacillus* (A_), and our genomes (LAT_11, LAT_26, LAT_27, LAT_31, LAT_33, LAT_35, and LAT_38). The colors distinguish the two main clades. (**b**) Phylogenomic tree (maximum likelihood estimation); the tree is colored by genus: yellow represents the genus *Geobacillus*, blue represents *Brevibacillus*, and green represents *Anoxybacillus*. Clusters with dotted lines correspond to the Antarctic *Anoxybacillus* strains isolated in the present study.

**Figure 3 microorganisms-10-01673-f003:**
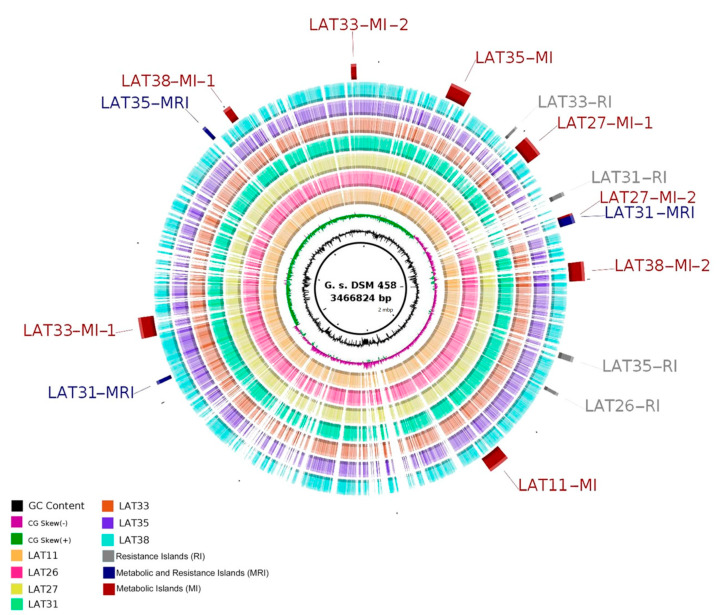
Circular visualization of the genome comparisons of *Anoxybacillus* strains and *Geobacillus stearothermophilus* DSM 458 (reference), created by BRIG. The inner black circle contains the complete reference genome (strain DSM 458), and the intensity of each color indicates the similarity of that strain with the reference genome.

**Figure 4 microorganisms-10-01673-f004:**
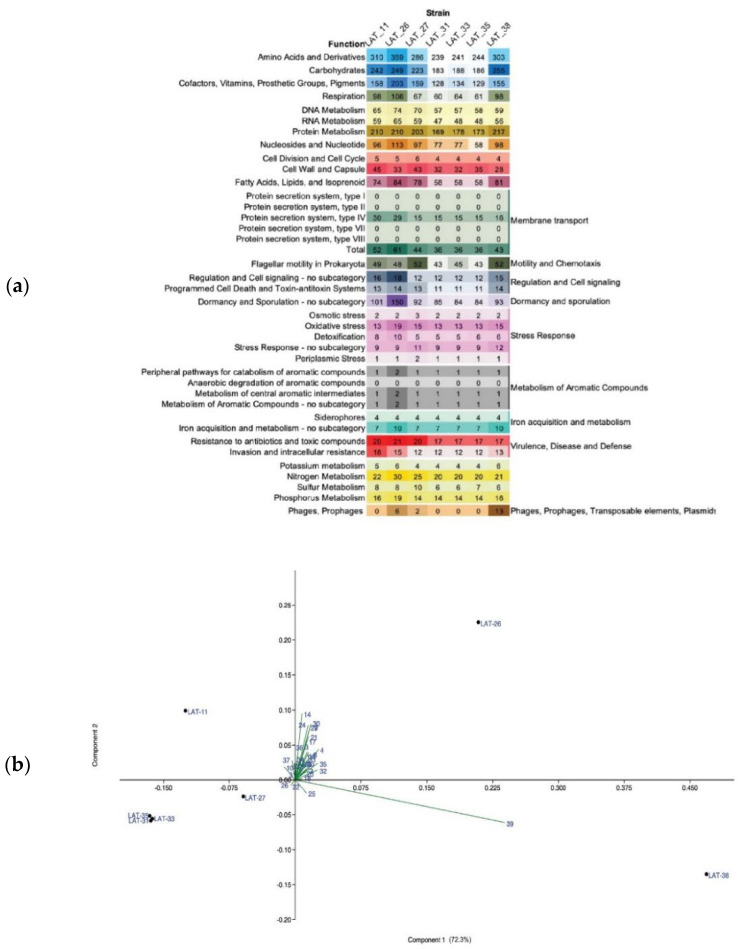
Functional analysis of *Anoxybacillus* strains described in the present study. (**a**) Number of genes related to metabolic functions in *Anoxybacillus* strains. (**b**) PCA plot of the annotated functional categories in *Anoxybacillus* strains; the *Anoxybacillus* strains are represented by dots, and the functional categories are represented by numbered green vectors. The angles and lengths of radiating vectors indicate the direction and strength of the relationships between the function and strain, respectively.

**Table 1 microorganisms-10-01673-t001:** Characteristics of the seven bacterial strains previously identified as belonging to the *Anoxybacillus* genus isolated from both geothermal sites on Deception Island, Antarctica.

*Anoxybacillus* Strain	Location	Environmental Temperature	Conditions of Isolation	Temperature Range of Growth
LAT_11	Fumarole Bay, Deception Island, Antarctica	100 °C	Lysogeny agar, 55 °C	30–70 °C
LAT_26	Fumarole Bay, Deception Island, Antarctica	55 °C	DSMZ 260 medium, 55 °C	30–70 °C
LAT_27	Fumarole Bay, Deception Island, Antarctica	70 °C	Lysogeny agar, 55 °C	30–70 °C
LAT_31	Fumarole Bay, Deception Island, Antarctica	70 °C	DSMZ 260 medium, 55 °C	30–70 °C
LAT_33	Fumarole Bay, Deception Island, Antarctica	70 °C	DSMZ 260 medium, 55 °C	30–70 °C
LAT_35	Fumarole Bay, Deception Island, Antarctica	80 °C	Lysogeny agar, 55 °C	30–70 °C
LAT_38	Whalers Bay, Deception Island, Antarctica	50 °C	Lysogeny agar, 55 °C	30–70 °C

**Table 2 microorganisms-10-01673-t002:** General genomic features of *Anoxybacillus* strains analyzed in the present study.

Strain	Genome Size (bp)	G + C Content (%)	Number of CDS	Count of RNA	Number of Contigs	Accession Number
LAT_11	3,296,946	41.8	3668	100	402	JAILSF000000000
LAT_26	3,572,987	42.0	3993	120	249	JAILSG000000000
LAT_27	3,160,393	42.0	3575	131	491	JAIWIK000000000
LAT_31	2,713,308	41.6	2885	100	102	JAILSH000000000
LAT_33	2,730,467	41.7	2932	109	123	JAILSI000000000
LAT_35	2,720,400	41.7	2931	102	129	JAILSJ000000000
LAT_38	3,368,374	43.5	4297	127	1122	JAILSK000000000

**Table 3 microorganisms-10-01673-t003:** CRISPR-Cas system subtypes found in the analyzed genomes.

Gene	System Subtype	Strain
CAS12f4_0	V-F	LAT_11, LAT_26, LAT_27, LAT_35, LAT_33, and LAT_31
CAS7_2	I-B	LAT_27
CSX1_7	III	LAT_27
CSB3_0	I-U	LAT_11
CSE1_0	I-E	LAT_38

## Data Availability

The complete genome sequence data, including raw sequence reads, genome assemblies, and annotations of *Anoxybacillus* strains LAT_11, LAT_26, LAT_27, LAT_31, LAT_33, LAT_35, and LAT_38 used in the present study were submitted to NCBI GenBank under the BioProject accession number PRJNA647943, available at https://www.ncbi.nlm.nih.gov/bioproject/PRJNA647943. The genomes used for comparison are available in Appendix A.

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
