# Peer review of "Unraveling the Genomic Potential of the Thermophilic Bacterium Anoxybacillus flavithermus from an Antarctic Geothermal Environment"

_microorganisms, 2022, doi:10.3390/microorganisms10081673_

Round 1

Reviewer 1 Report

The authors well reported the phylogenomics and potential functionalities of seven Anoxybacillus strains isolated from the sediment samples of Deception Island, an active polar volcano in Antarctic Peninsula, This is the first study to explore Anoxybacillus flavithermus species in the Antarctic geothermal environment. In the present study, The authors analyzed, for the first time, the presence, relative location, and gene composition of genomic islands (GIs) in the bacterial species A. flavithermus.  Moreover, since A. flavithermus strains are widely isolated from thermophilic environments, however, little is known about their presence and adaptations to thrive in geothermal ecosystems in polar regions. To overcome this gap, this study revealed the lifestyle of the relatively new genus Anoxybacillus, its adaptations to harsh environmental conditions of Antarctica, and its unique bioproducts that can have potential biotechnological applications, for example in starch- and lignocellulose-related industries.

In my opinion, in the manuscript the data and analyses are  presented appropriately, and it can be publish in Microorganisms in the present form.

Author Response

A:  We thank the reviewer for the positive feedback.

Reviewer 2 Report

The paper entitled The Chronicles of Ice and Fire: Unraveling the genomic potential of the thermophilic bacterium Anoxybacillus flavithermus in a geothermal environment in Antarctica have a main purpose to report the features, phylogenomics and potential functionalities of seven Anoxybacillus strains isolated from the sediment samples of Deception Island, an active polar volcano in a marine system, with continuous fumaroles and accentuated gradient of temperature, salinity and geochemistry located in the Antarctic Peninsula, based on the analysis of their genomes. The primary goals of the paper were (i) Knowledge discovery in databases: a systematic review of the literature and molecular databases, (ii) Sampling site, isolation, and culture conditions and (iii) DNA extraction and genome sequencing, assembly, and annotation; Phylogenomic analysis; Comparative genomic analyses.

The paper is structured in five parts excepting abstract and references. In part one, Introduction are presented the state of the art into field.

The part two is structured in five chapters in which are detailed the methods used for the obtaining the results according to the goals of the study. In part tree is described in detail the obtained results. This part is detailed in four subchapter in which are described and comment the results. Chapter four is dedicated to the discussion of the results into frame of the state of the art in the investigated field.

As conclusions, paper revealed that A. flavithermus strains are widely isolated from thermophilic environments; however, little is known about their presence and adaptations to thrive in geothermal ecosystems in polar regions. Using genomic analysis, the present study revealed, for the first time, the lifestyle of the relatively new genus Anoxybacillus, its adaptations to overcome the harsh environmental conditions of Antarctica, and its unique bioproducts that can have potential biotechnological applications.

Considering the quality of the manuscript I recommended to be accepted for publication without further modifications.

Author Response

A: We appreciated your review. Thank you for your positive feedback.

Reviewer 3 Report

The authors present a genome sequence analysis study of seven new Antarctic Anoxybacillus flavithermus isolates, including a phylogenomic comparison. The manuscript is well written and provides all appropriate annotations which are clearly presented in figures and tables. However, the manuscript lacks a deeper analysis that goes beyond listing numbers of genes in specific functions to fall within the scope of Microorganisms. In my opinion, it does not give sufficient insight in how the physiological potential of these strains compares to that of the other Anoxybacillus genomes and those of other species, which would better demonstrate what makes these strains special. To give an example, Figure 7 provides the number of genes for each isolate in the different functions. Based on this information one still does not know which of the amino acids the strains should be capable of making themselves or which carbohydrates it should be able to utilize and how that differs between the strains and between these strains and the other Aneurinibacillus species. Another example is the part on the adaptation to thermal stress. Here the authors list specific genes, but do not clearly state how unique it is for bacteria to have such combinations of genes, making it hard to see if the conclusion that these contribute to the adaptation of the strains to their harsh environment is justified. Finally, there is a tendency in overselling the potential for the biotechnological and astrobiological applications without clearly pointing out what is so unique about these strains (apart from referring to other studies). This way it remains highly speculative.

Specific comments:

Line 39 This sentence has a comparison to an unknown reference. "closer to each other[...] and to..." probably should read "closer to each other[...] than to..."

Line 78 "environemntal" should read "environmental"

Line 79 "extreme pH conditions" should be introduced/defined.

Lines 89-90. It remains unclear to me why specifically A. flavithermus should be explored in polar regions.

Line 131. "The equations should be inserted in editable format from the equation editor". I have no clue what the authors mean with this sentence.

Line 158 The authors comment about the growth temperature of 55C: "The chosen temperature was based on the in-situ temperature of both sampled geothermal sites." In Table 1 I read that the environmental temperature of only two out of seven strains were close to 55C; three were 70C, one 80C and one 100C. The authors do not show the temperature range for growth of these isolates, but for me it is unlikely that the strains will have been in their vegetative stage and alive at the sites of 70C and higher. Surprisingly, the authors do not comment on this.

Line 207 "pb" should read "bp"

Paragraph 3.3. In line with this study and all the analyses provided, I would have expected that the authors would also include a digital DNA-DNA hybridization analysis.

Line 334 "closer to" should read "closest to" to avoid having an incomplete comparison.

Lines 382-384. I have no clue what the authors mean to say with "For the sake of clarity, only the resistance and metabolic islands were predicted once a species was found to be non-pathogenic ad did not exhibit symbiosis with plants."

Lines 453-456. This is the only detail on carbohydrate metabolism that the authors provide apart from the number of genes involved listed in Figure 7. It would be nice if they list the ORF numbers in the supplementary materials to allow easy access.

Line 480 "the genomes" should read "several genomes" (or quantify the number of genomes that have been sequenced).

Lines 491-492 "Such studies are discussed in the Supplementary Table 6." The studies are not discussed, but listed in Table 6.

Lines 512-524. The authors should have a better look at the way they introduce CRISPR-Cas without mixing-up terms, and should explain all abbreviations (e.g., crRNA, PAM); Not all systems are Cascade; The PAM is not the only factor determining the recognition site; ...

Lines 527-544. High temperature tolerance is explained by the presence of a set of general stress response genes, but it is not clear if and how these link to high temperature tolerance and how unique it is for bacilli to have these genes. Are they specific to thermophiles?

Lines 580-582. For an in-depth analysis of carbohydrate metabolism I would expect a list with potential carbohydrate substrates. Not only a listing of the glycosyl hydrolases.

Lines 601-615. What distinguishes the A. flavithermus enzymes from those of other bacilli?

Author Response

Reviewer 3

The authors present a genome sequence analysis study of seven new Antarctic Anoxybacillus flavithermus isolates, including a phylogenomic comparison. The manuscript is well written and provides all appropriate annotations which are clearly presented in figures and tables. However, the manuscript lacks a deeper analysis that goes beyond listing numbers of genes in specific functions to fall within the scope of Microorganisms. In my opinion, it does not give sufficient insight in how the physiological potential of these strains compares to that of the other Anoxybacillus genomes and those of other species, which would better demonstrate what makes these strains special. To give an example, Figure 7 provides the number of genes for each isolate in the different functions. Based on this information one still does not know which of the amino acids the strains should be capable of making themselves or which carbohydrates it should be able to utilize and how that differs between the strains and between these strains and the other Aneurinibacillus species. Another example is the part on the adaptation to thermal stress. Here the authors list specific genes, but do not clearly state how unique it is for bacteria to have such combinations of genes, making it hard to see if the conclusion that these contribute to the adaptation of the strains to their harsh environment is justified. Finally, there is a tendency in overselling the potential for the biotechnological and astrobiological applications without clearly pointing out what is so unique about these strains (apart from referring to other studies). This way it remains highly speculative.

A: Thank you very much for your constructive comments. We have welcomed the suggestions to make certain changes and truly agree our manuscript has greatly benefited from your review. We have performed adjustments in all the sections, and we believe the manuscript is now much improved. Please see further details pertaining to each specific suggestion below.

Specific comments:

Line 39 This sentence has a comparison to an unknown reference. "closer to each other[...] and to..." probably should read "closer to each other[...] than to..."

A: Corrected.

Line 78 "environemntal" should read "environmental"

A: Corrected.

Line 79 "extreme pH conditions" should be introduced/defined.

A: We thank the reviewer for this suggestion. We modified the sentence in the revised manuscript: “[…] alkalophilic conditions (e.g., pH higher than 8.5)”.

Lines 89-90. It remains unclear to me why specifically A. flavithermus should be explored in polar regions.

A: Overall, the findings of Anoxybacillus genus is poorly reported; information regarding the species A. flavithermus were not previously described in an Antarctic volcano. The majority of information available in the literature is related to the presence of Anoxybacillus strains in dairy industries. To the best of our knowledge, this is the first study attempting to unveil the genomic features and metabolic potentials of A. flavithermus species in an Antarctic geothermal environment. We believe that A. flavithermus isolated from polar regions could be useful as candidates for potential applications in biotechnology and astrobiology, due to the harsh environmental conditions that they are thriving in, i.e., fluctuations in high and low temperatures at short distances, high UV irradiation, and in the presence of salt (due to the marine influence). Additionally, more published genomes are needed in the public databases aiming to contribute to other research studies and for further discoveries based on genomic analysis.

Line 131. "The equations should be inserted in editable format from the equation editor". I have no clue what the authors mean with this sentence.

A: Thank you for pointing this out. It was a typo. The paragraph should start with the sentence “Deception Island is characterized as a stratovolcano with a horseshoe shape […].” The sentence was corrected in the revised manuscript.

Line 158 The authors comment about the growth temperature of 55C: "The chosen temperature was based on the in-situ temperature of both sampled geothermal sites." In Table 1 I read that the environmental temperature of only two out of seven strains were close to 55C; three were 70C, one 80C and one 100C. The authors do not show the temperature range for growth of these isolates, but for me it is unlikely that the strains will have been in their vegetative stage and alive at the sites of 70C and higher. Surprisingly, the authors do not comment on this.

A: The temperature of 55°C was selected based on the temperature found in Fumarole Bay (ranging from 55 to 100°C) and Whalers Bay (from 50 to 60°C). This information was not written in the manuscript and we added for better clarity (lines 191-192, in the revised manuscript). Also, at the moment of the sampling, the in-situ temperature was measured, and the environmental temperature written in the Table 1 is related to the sampling site in Deception Island, Antarctica. The strains of A. flavithermus were isolated at 55°C in laboratory conditions, however, they came from the samples ranging from 50 to 100°C.

Regarding the range of temperature for the cultivation of the seven A. flavithermus species, we tested in a range of 30 to 100°C, and all the seven strains grew in a range of 30 to 70°C, corroborating with the related literature. This data was not shown in the manuscript, but it is available in the revised manuscript (Table 1).

Line 207 "pb" should read "bp"

A: Corrected.

Paragraph 3.3. In line with this study and all the analyses provided, I would have expected that the authors would also include a digital DNA-DNA hybridization analysis.

A: We have performed the digital DNA digital hybridization (dDDH) using the Genome-to-Genome Distance Calculator (GGDC 2.1) (Meier-Kolthoff et al. 2013) provided by Leibniz on the DSMZ Institute website (http://ggdc.dsmz.de/distcalc2.php). The text in the revised manuscript was modified in the material and methods section (lines 286-290) and in the results sections (paragraph 377-398). In case is needed, we can provide a supplementary table with the ANI, tetra and dDDH values obtained with these analyses.

Line 334 "closer to" should read "closest to" to avoid having an incomplete comparison.

A: Corrected.

Lines 382-384. I have no clue what the authors mean to say with "For the sake of clarity, only the resistance and metabolic islands were predicted once a species was found to be non-pathogenic ad did not exhibit symbiosis with plants."

A: We have removed this sentence from the revised manuscript.

Lines 453-456. This is the only detail on carbohydrate metabolism that the authors provide apart from the number of genes involved listed in Figure 7. It would be nice if they list the ORF numbers in the supplementary materials to allow easy access.

A: Based on your suggestion, we modified the revised manuscript. Regarding the information of annotated carbohydrates metabolism, we prepared a new supplementary material and it is available in the Supplementary table 8. Thank you for your comment.

Line 480 "the genomes" should read "several genomes" (or quantify the number of genomes that have been sequenced).

A: Corrected.

Lines 491-492 "Such studies are discussed in the Supplementary Table 6." The studies are not discussed, but listed in Table 6.

A: Thank you for pointing this out. We have corrected the sentence in the revised manuscript.

Lines 512-524. The authors should have a better look at the way they introduce CRISPR-Cas without mixing-up terms, and should explain all abbreviations (e.g., crRNA, PAM); Not all systems are Cascade; The PAM is not the only factor determining the recognition site; ...

A: In lines 515-527, the sentences were modified to improve the clarity, adding the meaning of the abbreviations: “CRISPR (clustered regularly interspaced short palindromic repeats) and CRISPR-associated (Cas) proteins is a defense system presented in microorganisms and acts against phages and others foreign genetic elements (Hille et al., 2018). The complex of Cas proteins (CRISPR-Cas) uses Cas proteins (Jore et al., 2011; Wiedenheft et al., 2011) linked to CRISPR RNA (crRNA). This complex is directed to the proper sequence present in the invading DNA usually through the recognition of the Protospacer Adjacent Motif (PAM) domain in the complementary strand of crRNA (Hille et al., 2018). We found genes belonging to class 1 (type I and III). Type I is the most widespread type in CRISPR systems (Makarova et al., 2015). In almost all type I systems, pre-crRNA is processed by an RNase of the Cas6 family. Type III has a peculiarity; it is divided into two subtypes. CRISPR type III-A is capable of acting on plasmid DNA in vivo, while CRISPR type III-B is responsible for the cleavage of only single-stranded RNA in vitro. We also found one gene belonging to class 2 (type V). Type V is distinguished by a single RNA-guided RuvC domain-containing effector, Cas12. However, additional studies are required to assess whether the predicted cassettes are rare subtypes of CRISPR-Cas systems and whether they are functional.

Lines 527-544. High temperature tolerance is explained by the presence of a set of general stress response genes, but it is not clear if and how these link to high temperature tolerance and how unique it is for bacilli to have these genes. Are they specific to thermophiles?

A: Thermophiles present a myriad of key genes and metabolisms to respond and overcome the thermal challenges. The majority of these genes are specific for thermophiles and thermotolerants, due to their cell adaptations over the time; they evolve to defeat the threats in the extremosphere. For better clarity, we have improved the text in the revised manuscript (modifications in the topics 3.4 and 4.1).

Lines 580-582. For an in-depth analysis of carbohydrate metabolism, I would expect a list with potential carbohydrate substrates. Not only a listing of the glycosyl hydrolases.

A: Thank you for pointing this out. We recognize that the carbohydrate metabolism provides a range of potential applications; for this article, we focused only in the group of glycosyl hydrolase enzymes.

Lines 601-615. What distinguishes the A. flavithermus enzymes from those of other bacilli?

A: It will depend of the bacilli (family, phylum) and the environment that they are flourishing, because the pressure of extreme conditions, evolution and horizontal transfer gene can affect and benefit the individuum and/or species in a customized way. We hope our study can help to build knowledge on potential applications and differences among another related genus.

Round 2

Reviewer 3 Report

The revised manuscript has clearly improved compared to the first submission. I understand that genome sequence data can have nearly indefinite options to analyse their content and the authors have to make a selection. Although I have the feeling that their analysis has potential for some more in-depth discussion, it is acceptable to me in its current version. There are just a few comments that I advise the authors to look into.

1. One aspect that has not been addressed to my full satisfaction is the isolation temperature. The information on it is clear, but what I advise the authors to comment on is that the isolation temperatures of at least two strains, LAT_11 and LAT_35 are outside the temperature range of growth. As one would not expect live planctonic cells at those sampling sites, it is most likely that here spores were sampled instead. Would this then count as cells isolated from such a site?

2. I agree with omitting former Figures 2, 4 and 5 from the body of the text. Why not adding these to the Supplementary Material?

3. I spotted the following typos (line numbers of track changes version):

Line 413 ...genes in carbohydrate metabolism were revealed...

Line 414 ...We focussed on the...

Line 424 ...the number of genes [...] was greater...

Line 445 ...presented in Figure 2.

Line 533 ...and CRISPR-associated (Cas) proteins are a defense system present in microorganisms and acting against phages...

Line 569 ...contained various genes encoding heat- and cold-shock proteins. OR ...contained various genes involved in heat- and cold-shock.

Line 619-621 Our strains exhibited genes related to adaptation to thermal fluctuation and alkaline pH (e.g.,  β-xylosidase, α-L-arabinofuranosidase, and β-glucosidase). I think this is not the right set of examples for these specific characteristics.

Author Response

We would like to thank again the reviewer 2 for the constructive comments of this second round of review. We have welcomed all the suggestions in the revised manuscript (microorganisms-1804505-2) and a point-by-point format is provided below, as well as you can find these changes marked in the revised manuscript.

Comments and Suggestions for Authors

The revised manuscript has clearly improved compared to the first submission. I understand that genome sequence data can have nearly indefinite options to analyse their content and the authors have to make a selection. Although I have the feeling that their analysis has potential for some more in-depth discussion, it is acceptable to me in its current version. There are just a few comments that I advise the authors to look into.

  1. One aspect that has not been addressed to my full satisfaction is the isolation temperature. The information on it is clear, but what I advise the authors to comment on is that the isolation temperatures of at least two strains, LAT_11 and LAT_35 are outside the temperature range of growth. As one would not expect live planctonic cells at those sampling sites, it is most likely that here spores were sampled instead. Would this then count as cells isolated from such a site?

A: Thank you for your comment. We have added a sentence in order to clarify for the reader (lines 214-215). The strains LAT_11 and LAT_35 were isolated using an incubation temperature of 55°C; in-situ temperature in the sampled geothermal site were 80 and 100°C. This only mean that the strains can survive at higher temperatures than the isolation temperature. We cannot assure that they were obtained initially as spores or vegetative cells, but yes, both cases will count as isolate bacteria from the sampled site.

  1. I agree with omitting former Figures 2, 4 and 5 from the body of the text. Why not adding these to the Supplementary Material?

A: Thank you for the suggestion. We have added those deleted figures in the Supplementary Material.

  1. I spotted the following typos (line numbers of track changes version):

Line 413 ...genes in carbohydrate metabolism were revealed...

Line 414 ...We focussed on the...

Line 424 ...the number of genes [...] was greater...

Line 445 ...presented in Figure 2.

Line 533 ...and CRISPR-associated (Cas) proteins are a defense system present in microorganisms and acting against phages...

Line 569 ...contained various genes encoding heat- and cold-shock proteins. OR ...contained various genes involved in heat- and cold-shock.

Line 619-621 Our strains exhibited genes related to adaptation to thermal fluctuation and alkaline pH (e.g.,  β-xylosidase, α-L-arabinofuranosidase, and β-glucosidase). I think this is not the right set of examples for these specific characteristics.

A: We are grateful for pointing out the typos. We have corrected in the revised manuscript.